# Histological Kidney Re-Evaluation after Daratumumab Monotherapy for AL Amyloidosis

Roberta Fenoglio [1,*], Gianluca Rabajoli [1], Antonella Barreca [2], Emanuele De Simone [1], Savino Sciascia [1] and Dario Roccatello [1]

1   University Center of Excellence on Nephrologic, Rheumatologic and Rare Diseases (ERK-net, ERN-Reconnect and RITA-ERN Member) with Nephrology and Dialysis Unit and Center of Immuno-Rheumatology and Rare Diseases (CMID), Coordinating Center of the Interregional Network for Rare Diseases of Piedmont and Aosta Valley (North-West Italy), San Giovanni Bosco Hub Hospital, and Department of Clinical and Biological Sciences of the University of Turin, 10154 Turin, Italy; gianluca.rabajoli@gmail.com (G.R.); ema.desimone@gmail.com (E.D.S.); savino.sciascia@unito.it (S.S.); dario.roccatello@unito.it (D.R.)
2   Division of Pathology, Città della Salute e della Scienza Hospital, University of Turin, 10154 Turin, Italy; antonella.barreca@libero.it
*   Correspondence: roberta.fenoglio@unito.it

**Abstract:** Background: AL amyloidosis is a systemic disorder characterized by extracellular deposition of characteristic fibrils that results in progressive multi-organ failure and premature death. Recently daratumumab has been demonstrating higher hematological and organ response rates when compared to the standard of care. We hereby report our long-term experience on the effects of daratumumab given alone on the deposition of amyloid as evaluated in repeat renal biopsy. Results: Six patients were enrolled. All patients had proteinuria that was associated with renal function impairment in four. After therapy with daratumumab, four patients achieved complete hematological response and two had partial hematological response at the end of treatment. With regard to renal response, four out of six patients achieved an organ response; one patient had fluctuating proteinuria levels and did not meet the needed criteria at the end of the treatment and the last patient, who was already in dialysis at the time of therapy initiation, remained on dialysis despite complete hematological and cardiac responses. A significant decrease in 24-h proteinuria from 7.9 g/24 h to 1.1 ($p < 0.005$) with stabilization or improvement of sCr (from 1.5 mg/dL to 1.2 mg/dL; $p = 0.34$) were observed. All patients underwent a repeat biopsy after 24 administrations of daratumumab. In five patients, the repeat biopsy showed unchanged features; while in one it showed an improvement. Conclusions: Our data, based on real life experience, show that daratumumab monotherapy can be an effective therapeutic option. It is capable not only of achieving a substantial rate of renal improvement in pre-treated and naïve patients, but also of limiting renal deposition

**Keywords:** amyloidosis; daratumumab; repeat renal biopsy; amyloid deposits; NT-proBNP



## 1. Introduction

AL amyloidosis is a systemic disorder characterized by extracellular deposition in virtually every tissue of beta amyloid fibrils, deriving from a pathological misfolding of IG monoclonal free light chains (FLCs), which results in progressive multi-organ failure and premature death [1].

Even if progress has been made over the last two decades, both in terms of therapeutic options and early diagnosis, prognosis continues to be extremely poor for the affected patients [2].

Current gold standard therapy protocols are based on autologous stem cell transplantation (ASCT), but less than 50% of patients are eligible at diagnosis, mostly because of their disease-related frailty [3,4]. For those who are not eligible for ASCT, baseline therapeutic schemes are derived from multiple myeloma protocols and include proteasome inhibitors,

alkylating agents, immunomodulatory drugs (IMiD) and steroids, with Bortezomib-based regimens being the most commonly used [5]. However complete hematological response rates (the best predictor of a good clinical outcome) remain unsatisfactory.

Our previous experience [6] with conventional therapy with bortezomib-based conventional therapy in systemic AL amyloidosis with renal involvement was highly disappointing. While two-thirds of patients experienced a hematologic response, only one-third of 20 patients showed some renal improvement. Notably, the majority of repeat biopsies in patients undergoing conventional therapy showed increased amyloid deposits suggesting that management with conventional therapy is unable to influence the process of amyloid deposition.

The recently concluded ANDROMEDA trial [7] led to the approval of daratumumab, a first-in-class anti-CD38 human antibody (IgG1κ), for the treatment of AL amyloidosis in association with VCD (cyclophosphamide, bortezomib, dexamethasone). This trial demonstrated higher hematological and organ response rates and a satisfactory safety profile when compared to the standard of care.

The rationale for the use of daratumumab in AL amyloidosis is strong: it targets CD38, which is highly expressed by pathogenic plasmacells (PCs), leading to their death via multiple mechanisms of apoptosis and killing of the cells responsible for the production of affected FLCs and, therefore, interrupting amyloid deposition [8,9]. However, in that trial daratumumab was tested as an add-on therapy, was associated with significant drug-related toxicity (mostly due to Bortezomib and steroids) and could not be evaluated for its individual efficacy.

Our group previously reported an initial experience with daratumumab monotherapy in AL amyloidosis in a small series of patients with severe biopsy-proven renal involvement [10]. We hereby report our long-term experience on the effects of a complete course (i.e., 24 administrations) of daratumumab given alone on the deposition of amyloid as evaluated in repeat renal biopsy.

## 2. Materials and Methods

We selected patients affected by AL amyloidosis who underwent a repeat renal biopsy after a course of 24 administrations with daratumumab monotherapy. Diagnosis was based on a combination of serological tests and histological analysis: apart from renal biopsy, all patients underwent bone marrow biopsy and were evaluated for peripheral nervous system involvement by electromyography, relief of orthostatic pressure and cardiac involvement as assessed by echocardiography and the measure of serum levels of NT-proBNP. Renal function was assessed by measuring eGFR in accordance with the Kidney Disease Outcomes Quality Initiative guidelines using the formula of Modification of Diet in Renal Disease (MDRD) and detecting proteinuria obtained by 24-h urine collection.

At the time of daratumumab monotherapy initiation, all patients had severe biopsy-proven renal involvement. The treatment protocol included 16 mg/kg daratumumab administered intravenously weekly for eight consecutive weeks, then every two weeks eight more times and lastly monthly until the 52nd week. Premedication included paracetamol 1000 mg (oral), chlorphenamine 10 mg (intravenous) and methylprednisolone 125 mg (intravenous).

Daratumumab was obtained by the local hospital pharmacy for off-label use, according to the rules for the management of Rare Diseases of Piedmont (north-west Italy). All patients provided informed consent.

Renal response was defined according to the International Society of Amyloidosis extended criteria as a ≥30% decrease in proteinuria or drop of proteinuria below 0.5 g/24 h in the absence of renal progression (≥25% decrease in eGFR) [11].

Clinical response was defined as an improvement in organ function or performance status. All patients were evaluated after the 4th, 8th, 16th and 24th infusion and then every three months by examining hematological response, biomarkers of involved organs

and clinical complications. All patients completed the 24th infusion of daratumumab and underwent repeat renal biopsy.

Biopsy specimens were examined using light microscopy (LM) and immunofluorescence (IF). Kidney tissue sections were evaluated by hematoxylin and eosin, periodic acid-Schiff (PAS) and Masson's trichrome stains. Staining with Congo red was used to confirm amyloid deposits. Light chain amyloidosis (AL) was identified by light chain restriction using immunohistochemistry.

### 3. Results

Six patients were enrolled (two males and four females), mean age at diagnosis was 68.4 years (range 56–81 years) (Table 1). Initial results of four out of six patients were included in a previous report [10]. Their data in the present cohort refer to a longer follow-up period than that originally reported, and the results of repeat biopsy after treatment. Four out of six patients had undergone previous treatments, the other two patients were treated with daratumumab as frontline treatment.

**Table 1.** Demographic and clinical characteristics of patients at baseline.

|  | Sex | Age | M Component | Previous Treatment | K/λ RATIO | Bence Jones | NT-ProBNP pg/mL | Serum Cr (mg/dl) | UPt (g/day) |
|---|---|---|---|---|---|---|---|---|---|
| PT 1 | M | 60 | IgAλ | B-CYC-D M-D | 0.02 | ++ | 8296 | HD | HD |
| PT 2 | F | 72 | IgGλ | L-CYC-D | 0.21 | ++ | 342 | 0.6 | 2.5 |
| PT 3 | F | 56 | λ | M-D | 0.23 | ++ | 1840 | 0.8 | 6.8 |
| PT 4 | M | 70 | λ | – | 0.06 | + | 168 | 2.4 | 9.3 |
| PT 5 | F | 81 | IgMλ | B-RTX-D | 0.79 | + | 1727 | 2.5 | 5.5 |
| PT 6 | F | 70 | IgAk | – | 1.18 | + | 222 | 3 | 7.4 |

B: bortezomib, CYC: cyclophosphamide, D: dexamethasone, M: melphalan, L: lenalidomide, RTX: rituximab, sCr: creatinine, UPT: proteinuria, HD: hemodyalisis, NT-proBNP.

Three out of six patients had cardiac involvement (mean NT-proBNP was 2,099.17 ng/L) with two of them having a Mayo stage 3b disease (i.e., NT-proBNP > 1800 ng/L). Three out of six had peripheral nervous system involvement and two of six had gastrointestinal involvement. Serum FLC ratio was altered at the time of diagnosis in four of six. All patients had proteinuria (in the nephrotic range in 5) that was associated with renal function impairment in four. One patient was on dialysis at the time of therapy initiation. Extensive amyloid renal infiltration was found in all patients. Four out of six patients had a diffuse pattern characterized by vascular, glomerular and interstitial amyloid deposition. One patient had vascular and glomerular amyloid deposition and one had glomerular and interstitial deposition. Marked positivity (+++) for the substance P was observed in all the biopsies.

After therapy with daratumumab, four patients achieved complete hematological response and two had partial hematological response after eight infusions and at the end of treatment. All patients with cardiac involvement obtained resolution or amelioration. With regard to renal response, five out of six patients achieved an organ response, but one patient had fluctuating proteinuria levels and at the end of the treatment did not meet the required criteria for renal response; the remaining patient, who was in dialysis at the time of therapy initiation, remained on dialysis despite complete hematological and cardiac responses. All patients underwent a repeat biopsy after 24 administrations of daratumumab (Table 2).

A significant decrease in 24-h proteinuria from 7.9 g/24 h (range 2.5–9.3) to 1.1 (range 0.3–2.6 gr/die, $p < 0.005$) with stabilization or improvement of sCr (from 1.5 mg/dL to 1.2 mg/dL, $p = 0.34$) were observed (Tables 3 and 4).

Daratumumab was generally well tolerated. One patient experienced an infusion reaction during the first dose (grade 1). Thanks to pre-medication, no other adverse events occurred.

**Table 2.** Distribution of amyloid deposits in the kidney before and after treatment with daratumumab.

| | Glomerular Involvement Pre/Post | Interstitial Involvement Pre/Post | Vascular Involvement Pre/Post |
|---|---|---|---|
| Pt 1 * | +/++ | ++/++ | +/+ |
| Pt 2 ** | +++/++ * | - | +/+ |
| Pt 3 | ++/+ | +/+ | ++/+ |
| Pt 4 | ++/++ | ++/++ | +/+ |
| Pt 5 | +++/+++ | ++/++ | ++/++ |
| Pt 6 ** | ++/++ | +/+ | + |

* Patient #1 underwent dialysis, ** Patient #2 and #6 achieved a partial renal response.

**Table 3.** Proteinuria values of all patients all of the time during the treatment. TO: pre-treatment, and every other four infusions.

| UPT g/Day | PT 1 | PT 2 | PT 3 | PT 4 | PT 5 | PT 6 |
|---|---|---|---|---|---|---|
| T0 | HD | 1.80 | 6.80 | 9.30 | 5.50 | 7.40 |
| T4 | HD | 1.50 | 4.40 | 8.20 | 2.60 | 3.7 |
| T8 | HD | 1.30 | 4.20 | 3.00 | 1.90 | 3.3 |
| T12 | HD | 1.10 | 4.30 | 3.10 | 1.30 | 2.50 |
| T16 | HD | 2.10 | 2.70 | 2.20 | 1.40 | 2.90 |
| T20 | HD | 0.97 | 1.20 | 2.00 | 1.40 | 2.30 |
| T24 | HD | 1.60 | 0.30 | 1.90 | 0.90 | 2.60 |

**Table 4.** Serum creatinine values of all patients all of the time during the treatment. TO: pre-treatment, and every other four infusions.

| sCr mg/dL | PT 1 | PT 2 | PT 3 | PT 4 | PT 5 | Pt 6 |
|---|---|---|---|---|---|---|
| T0 | HD | 0.60 | 0.8 | 2.40 | 2.50 | 1.30 |
| T4 | HD | 0.70 | 0.8 | 2.40 | 1.80 | 1.00 |
| T8 | HD | 0.60 | 0.7 | 2.20 | 1.90 | 0.90 |
| T12 | HD | 0.50 | 0.7 | 2.80 | 1.90 | 1.20 |
| T16 | HD | 0.70 | 0.8 | 2.40 | 1.80 | 1 |
| T20 | HD | 0.50 | 0.7 | 2.30 | 1.90 | 1.2 |
| T24 | HD | 0.60 | 0.8 | 1.80 | 1.80 | 1.30 |

At the end of follow-up (mean 30.3 months, range 23–39), three patients have persistent hematological and renal response and one patient has persistent partial response. The second patient with initial partial response had a relapse and initiated a treatment with Bortezomib plus cyclophosphamide and dexamethasone. The last patient, who was on dialysis, died after 24 months of follow-up due to COVID-19 infection. At that time he had persistent hematological response.

## 4. Discussion

The optimal management of patients with AL amyloidosis remains to be defined. In particular, patients who are ineligible for transplant continue to have a poor outcome with a median overall survival of less than one year in those with cardiac involvement [12]. Since 2003, several studies have shown that treatment reducing the concentration of the amyloidogenic FLCs and improving cardiac dysfunction results in prolonged survival [2]. In recent years daratumumab has emerged as an appealing therapeutic alternative as shown by several reports [13–16]. Initial results were confirmed by the phase 3 ANDROMEDA study [7], in which daratumumab allowed to obtain higher frequencies of hematologic

complete response and survival free from major organ deterioration. However, in that trial daratumumab was added to bortezomib, cyclophosphamide and dexamethasone. The aim of treatment in AL amyloidosis is to eliminate fibril precursor proteins by suppressing the production of FLCs to rescue organ function. An attempt was made to propose a scoring system [17]. However, data concerning renal histological changes after therapy are lacking. Currently, renal response criteria are defined as a 30% decrease in 24-h urine protein that must be >0.5 g/day pretreatment, and creatinine or creatinine clearance that must not worsen by more than 25% over baseline [11]. Patients' outcome after daratumumab monotherapy was generally good. Stable estimated GFR and significant reduction in proteinuria were achieved in all but one patient at the end of follow up. This extended experience confirms our initial results and further supports the role of daratumumab monotherapy in the treatment of AL amyloidosis with severe renal involvement. As previously reported [6], a satisfying hematological response (meant as very good partial hematological response or complete hematological response [11]) is a prerequisite for achieving a renal response. Of the two patients showing a partial hematological response, only partial renal response was obtained. The patient that did not obtain a renal response was one of the two who had achieved only a partial hematological response.

To our knowledge, the present study is the first in which patients systematically underwent a repeat biopsy to re-evaluate renal damage. Amyloid deposits resulted in stability or improvement in all patients. Amyloid deposits were unchanged even in the patient who underwent dialysis, but there was a significant increase in sclerotic glomeruli and interstitial fibrosis. Notably, this patient had refractory disease and daratumumab was given as third-line therapy. He underwent three renal biopsies. The second biopsy, which was carried out before starting treatment with daratumumab, showed increased amyloid deposits, while the last one (after daratumumab treatment intended to limit cardiac impairment) showed a stabilization of amyloid deposits.

## 5. Conclusions

The present study describes six patients with AL amyloidosis and severe renal involvement who underwent a repeat biopsy after daratumumab monotherapy. This contribution is unique in the literature, and sets the scene for large-scale studies to further investigate histological aspects of renal damage. Our data, based on real life experience, show that daratumumab monotherapy can be an effective therapeutic option. It is capable not only of achieving a substantial rate of renal improvement in pre-treated and naïve patients, but also of limiting renal deposition. The treatment should be started as soon as possible, i.e., when amyloid deposits are limited, in order to anticipate the irreversible functional sequelae of extensive amyloid deposition.

**Author Contributions:** Conceptualization: R.F. and D.R., Resources: E.D.S. and A.B., Data collection: G.R. and R.F., Writing original draft preparation: R.F. and D.R. Supervision: S.S. All authors have read and agreed to the published version of the manuscript.

**Funding:** This research received no external funding.

**Informed Consent Statement:** Informed consent was obtained from all subjects involved in the study.

**Conflicts of Interest:** The authors declare no conflict of interest.

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
