# Peer review of "Histological Kidney Re-Evaluation after Daratumumab Monotherapy for AL Amyloidosis"

_hemato, doi:10.3390/hemato3020025_

Round 1

Reviewer 1 Report

The manuscript is fine, although lack of novelty and the result is well predicted as CD38 antibody already been approved for AL treatment, I am ok with its publication in Hemato.

A few minor comments:

For the only patient who showed improvement in biopsy sample: which patient? Did he or she showed complete hematological response to the CD38 antibody (CR or PR)? The authors also need to show the detail of the renal response of all the patients (e.g., protein urine) of all the time points during the treatment.

Figure 1 to 3, please indicate the amyloid deposits using arrows.

Author Response

Reviewer 1

The manuscript is fine, although lack of novelty and the result is well predicted as CD38 antibody already been approved for AL treatment, I am ok with its publication in Hemato.

Thanks for your comments. We believe that the novelties brought by this paper are the use of daratumumab alone with a relatively long subsequent follow-up and the availability of renal histological data.

A few minor comments:

For the only patient who showed improvement in biopsy sample: which patient? Did he or she showed complete hematological response to the CD38 antibody (CR or PR)?

The patient who showed improvement in biopsy sample is #pt3; she had a complete hematological response.

The authors also need to show the detail of the renal response of all the patients (e.g., protein urine) of all the time points during the treatment.

We added 2 tables (Table 3,4) with the detail of the renal response of all patients of all the time during the treatment as you suggested

Figure 1 to 3, please indicate the amyloid deposits using arrows.

The figures have been removed in accordance with the suggestion of another reviewer to omit the sentences about amyloid amount before and after treatment

Reviewer 2 Report

The anti CD38 antibody daratumumab has been shown to be an effective component in the of systemic AL amyloidosis. In this paper, this antibody was used as monotherapy in 6 patients with AL amyloidosis in whom renal involvement was a main manifestation. Renal function was evaluated and in addition, renal biopsy was performed both before and after the treatment period. Four patients achieved complete and two partial hematological response. Five patients obtained different degree of renal response. The amount of amyloid in the renal biopsies was evaluated to be unchanged in five patients and to be decreased in one.

Major points

  1. This is a remarkably good effect of monotherapy with daratumumab but the patient number is very low and further studies in larger series are necessary, something that the authors also claim. The authors’ conclusion that the amount of amyloid was stable or even reduced in the repeated renal biopsies seems shaky, however. I would strongly suggest to omit this part of the paper of several reasons. Firstly, the material is too small. Secondly, a much more careful measurement of the biopsies is necessary for this statement. Thirdly, it is very difficult to see from the figures what the authors claim. Therefore, also the last sentence of the Abstract should be changed.
  2. Why are the magnifications different in the Figures 1, 2, 3 A and B from those in C and D? This makes a comparison more difficult.

Minor

  1. Abstract, 1st sentence: ‘extracellular of beta amyloid fibrils’ is confusing. Beta amyloid can also be confused with the protein in Alzheimer’s disease. I suggest something like ‘extracellular deposition of characteristic fibrils’ or something similar.
  2. Please use consistently Daratumumab or daratumumab.

Author Response

Major points

This is a remarkably good effect of monotherapy with daratumumab but the patient number is very low and further studies in larger series are necessary, something that the authors also claim. The authors’ conclusion that the amount of amyloid was stable or even reduced in the repeated renal biopsies seems shaky, however. I would strongly suggest to omit this part of the paper of several reasons. Firstly, the material is too small. Secondly, a much more careful measurement of the biopsies is necessary for this statement. Thirdly, it is very difficult to see from the figures what the authors claim. Therefore, also the last sentence of the Abstract should be changed.

No question about the relatively small sample of patients under study. We omit the part of the paper as suggested by the reviewer in the conclusions and in the abstract

1. Why are the magnifications different in the Figures 1, 2, 3 A and B from those in C and D? This makes a comparison more difficult.

The figures have been removed in accordance with the suggestion to omit the sentences about amyloid amount before and after treatment

Minor

1. Abstract, 1st sentence: ‘extracellular of beta amyloid fibrils’ is confusing. Beta amyloid can also be confused with the protein in Alzheimer’s disease. I suggest something like ‘extracellular deposition of characteristic fibrils’ or something similar.

2. The sentence has been corrected as you suggested

3. Please use consistently Daratumumab or daratumumab.

Daratumumab has been changed in daratumumab

Round 2

Reviewer 2 Report

The authors have responded well to my criticism

Author Response

The  reviewer answered that we have responded well to his criticism. nothing to add